# Lessons Learned from 30 Years of Transverse Myocutaneous Gracilis Flap Breast Reconstruction: Historical Appraisal and Review of the Present Literature and 300 Cases

**DOI:** 10.3390/jcm10163629

**Published:** 2021-08-17

**Authors:** Laurenz Weitgasser, Maximilian Mahrhofer, Karl Schwaiger, Kathrin Bachleitner, Elisabeth Russe, Gottfried Wechselberger, Thomas Schoeller

**Affiliations:** 1Department of Plastic and Reconstructive Surgery, Marienhospital Stuttgart, Teaching Hospital of the Eberhard Karls University, 72074 Tübingen, Germany; maximilian.mahrhofer@vinzenz.de (M.M.); kathrin.bachleitner@vinzenz.de (K.B.); thomas.schoeller@vinzenz.de (T.S.); 2Department of Plastic and Reconstructive Surgery, Hospital of the Brothers of St. John of God (Barmherzige Brüder), Paracelsus Medical University, 5020 Salzburg, Austria; karl.schwaiger@bbsalz.at (K.S.); elisabeth.russe@bbsalz.at (E.R.); gottfried.wechselberger@bbsalz.at (G.W.)

**Keywords:** TMG flap, TUG flap, transverse myocutaneous gracilis flap, transverse upper gracilis flap, breast reconstruction, autologous breast reconstruction, microsurgery, free flap, free tissue transfer

## Abstract

Background: Happy 30th birthday to the transverse myocutaneous gracilis (TMG) flap. Since 1991 the TMG flap has been used to reconstruct a wide variety of defects and became a workhorse flap and reliable alternative to the deep inferior epigastric perforator (DIEP) flap in many breast reconstruction services worldwide. This manuscript sheds light on the history and success of the TMG flap by critically reviewing the present literature and a series of 300 patients receiving a breast reconstruction. Patients and Methods: The present literature and history of the TMG flap was reviewed and a retrospective double center cohort study of 300 free TMG free flaps for autologous breast reconstruction was conducted. Patient demographics, perioperative data, and post-operative complications were recorded and compared with literature findings. Results: Mean flap weight was 320 g. Mean pedicle length was 70 mm. Complications included 19 (6.3%) flap loss. 10 patients (3.3%) had postoperative cellulitis and 28 (9.3%) wound healing disturbance of the donor site. Conclusion: Recipient and donor site complications were comparable to other free flaps used for breast reconstruction. A low BMI or the lack of an abdominal based donor site do not represent a limitation for breast reconstruction and can be overcome using the TMG flap.

## 1. Introduction

Yousif et al. first published the detailed anatomy of the transverse myocutaneous gracilis (TMG) flap and clinical applications in the *Annals of Plastic Surgery Journal* in December 1991 [1]. However, due to problems of partial skin paddle necrosis with the gracilis myocutaneous flap it did not receive as much attention in the following years [2,3]. It was not until a series of publications by the senior authors of this study, Gottfried Wechselberger and Thomas Schoeller, between 2001 and 2004 that the TMG flap suddenly became a popular second option to the deep inferior epigastric perforator (DIEP) flap for autologous breast reconstructions [4,5,6]. In patients who do not offer an adequate lower abdominal donor site due to prior abdominal surgeries or a lack of soft tissue, the TMG flap can often be used alternatively due to the presence of excess soft tissue in the upper thigh. It was found that even in very athletic and low Body Mass Index (BMI) patients the TMG flap most often offers enough soft tissue for the autologous reconstruction of a small cup A to B breast.

Consequently numerous studies have further elaborated the TMG flap and the surgical techniques have been extensively refined further. Thereby offering save and reliable reconstructive results together with a low donor site morbidity comparable to abdominally-based flaps [7,8,9,10].With increasing experience the TMG flap became a powerful solution for immediate as well as delayed secondary and tertiary, uni- and bilateral breast reconstruction purposes [11].

In the last three decades countless patients worldwide have now benefited from this versatile and innovative myocutaneous flap and it represents a very well established option for breast reconstruction with known limitations in terms of volume and pedicle length.

Since the introduction of the fasciocutaneous profunda artery perforator flap in 2012 by Allen et al. which uses a similar angiosome for breast reconstruction, the TMG flap has faded from the spotlight a bit, but still represents an easy to raise and reliable workhorse flap in many breast reconstructive services [12,13]. This can be explained by the constant anatomy and relatively easy and convenient flap harvest compared to the PAP flap and the familiarity of many plastic surgeons with the gracilis flap itself. However, sacrifice of one of the adductor muscles poses a noteworthy drawback in the modern era of perforator based flap reconstructions.

This manuscript pays tribute to the 30th anniversary and summarizes the history and discusses the success of the TMG-flap. Moreover, a historical appraisal a review of the present literature focusing on the TMG flap for the use of breast reconstruction was conducted and 300 consecutive cases were retrospectively analyzed. Results were compared with literature findings of alternative popular free flap options to evaluate the current role of the TMG flap for autologous breast reconstruction today.

## 2. Patients and Methods

The present available literature from the first publication in the year 1991 until today was analyzed and the history and evolution of the TMG flap through this time period was critically evaluated. The senior authors (Gottfried Wechselberger and Thomas Schoeller) who were among the forerunners in TMG flap breast reconstruction back in the late 1990s and published numerous important manuscripts about its use share their knowledge and were asked to critically judge and discuss the current role of the TMG flap compared to its alternatives. Three hundred patients who had a unilateral breast reconstruction with a TMG flap at the senior authors institutions between September 2010 and October 2020 were reviewed in a retrospective double center cohort study. Detailed patient and flap characteristics as well as complications were categorized and recorded. Complications were divided into major and minor complications according to a modified classification system of Neaman et al. (Table 1) [14].

All patients were treated according to a standardized two-team approach of simultaneous flap harvest and recipient site dissection. An optimized surgical technique has been published recently [15]. Both senior authors have implemented identical pre-, intra- and postoperative protocols in each respective unit thus offering a large group of patients without compromising the data set. Data were checked for consistency in terms of typing errors, and ranges were inspected for validity. The study was conducted in accordance with the Ethical Principles for Medical Research involving Human Subjects of the Declaration of Helsinki. Intraoperative surgical technique of a unilateral breast reconstruction using the TMG flap is shown in Figure 1, Figure 2, Figure 3 and Figure 4. Pre- and postoperative images after completion of two-stage bilateral reconstruction are shown in Figure 5, Figure 6 and Figure 7.

## 3. Results

### 3.1. Patient Characteristics

Mean patient age was 48 years (18–77 range); 19.3% (58/300) of patients received a primary, 41.3% (124/300) a secondary, and 39.4% (118/300) a tertiary unilateral breast reconstruction. 38.6% (116/300) of patients underwent preoperative radiotherapy. Mean BMI was 23 kg/m^2^ (*SD* 3.1). Mean follow up time was 21.4 months (*SD* 20.9). 55% (165/300) of patients received postoperative lipofilling to improve breast symmetry. Detailed patient characteristics are listed in Table 2.

### 3.2. Flap Characteristics

The mean flap weight was 320 g (155–600 g range). The mean skin island diameter was 9 cm (7–13 cm range) in width and 31 cm (25 to 36 cm range) in length. Anastomosis to the mammary artery and vein was feasible in 92% of patients. The mean size of the venous coupler used was 2.5 mm (1.5 to 3.5 range). The mean pedicle length was 70 mm (43 mm to 110 mm range). Overall, 8.0% (24/300) of flaps needed to be anastomosed to the thoracodorsal artery and vein after failed or insufficient mammary artery anastomosis. Detailed flap characteristics are listed in Table 3.

### 3.3. Overall Complications

Overall complication rate was calculated to be 49% (147/300); 51% (153/300) of patients had no complication, 27% (83/300) had a major complication and 23% (70/300) had a minor complication (minor and major complications were assessed separately, double count possible). Flap take-backs to theatre due to venous or arterial insufficiency were recorded in 14.0% of patients (42/300). Flap loss was observed in 6.3% (19/300) of patients. Detailed complications are listed in Table 4.

### 3.4. Recipient Site Complications

Forty four patients (14.6%) had a hematoseroma of the recipient site; 15 patients (5.0%) demonstrated a postoperative fat necrosis; 16 patients (5.3%) suffered a cellulitis of the recipient site postoperatively; 16 patients (5.3%) had a wound healing disturbance of the recipient site which needed surgical management; 15 patients (5.0%) had to undergo later revision to remove a fat necrosis.

### 3.5. Donor Site Complications

Twenty one (7.0%) patients had a hematoseroma of the donor site; 10 (3.3%) patients had a cellulitis of the donor site which needed surgical management; 28 (9.3%) patients had a wound healing disturbance of the donor site which needed surgical management; 3 (1.0%) patients needed scar revision of the donor site. No patients had pain from a cluneal nerve neuroma postoperatively. No lymphocele or lymphedema was observed postoperatively in 300 reviewed flaps.

## 4. Discussion

The history of the TMG flap started in 1991 and it offers some controversy in terms of its denomination and popularity. After being first described by Yousif et al. it seems that surgeons and scientists were not aware of its potential at first and for the next 10 years no further studies on this interesting new flap were published [1]. The senior authors of this study started using it for breast-, head- and neck as well as extremity reconstruction and published their preliminary experience between 2001 and 2002 [6]. This sparked further interest in the TMG flap and more studies were published by Hallock and Arnez et al. in 2004 [7,16].

While the oldest studies used the name TMG flap, Arnez et al. came up with the name transverse upper gracilis (TUG) flap instead which led to utilization of both names in the surgical and scientific community. A current literature search in PUBMED is able to identify 35 studies using the name TMG flap while 40 studies use TUG flap today. This disparity was pointed out by Georg Huemer in 2013 who recommended a uniform denomination using TMG to improve future scientific communications, comparisons and referencing [17]. Besides the up to date present controversy regarding the flap’s name, it is also remarkable that a review of the current literature in PUBMED only revealed a total of 75 studies when searching for TMG or TUG flap. A graph listing the number of publications on the TMG flap between 2004 and today is shown in Figure 8.

In comparison, a total of 1251 studies on the DIEP flap could be identified. Although the TMG flap still serves as a popular secondary, and in some patients even primary, option for breast reconstruction this discrepancy points out a certain lack of research interest in the TMG flap. This can be explained in many ways. Compared to the DIEP flap which offers a variable angiosome depending on the perforator location and quality the TMG flap offers a constant angiosome and vascular anatomy which makes it a less exciting flap to study, but also a simple and reliable solution for breast reconstruction [5,18]. Numerous studies evaluated the complexity of the DIEP flap angiosome and were able to decode its variability depending on a lateral or medial perforator row location [19]. Similar to venous supercharging in DIEP flaps the TMG flap can be supercharged using the distal end of the great saphenous vein [5,20]. We found supercharging of TMG flaps rarely necessary since flap sizes are lower compared to DIEP flaps where supercharging is often indicated in higher flap weights [21].

Furthermore the TMG flap is competing with its sibling, the PAP flap, which was introduced by Allen et al. in 2012 and which can be offered to the same patient collective [7,12,22]. Since its introduction, the benefits of the PAP flap in comparison to the TMG flap have been under debate. While their angiosome on the inner upper posterior thigh as well as their weight (between 250 g and 450 g) are relatively similar, the PAP flap offers a longer pedicle length (70 to 150 mm vs. 60 to 80 cm) and potentially better vascular caliber match (average artery size 2.2 mm, average vein size 2.3 mm vs. 2.1 and 2.0 mm) to the mammary recipient vessels [23]. In comparison to the PAP, the much shorter pedicle of the TMG flap can make the anastomosis as well as postoperative revision of the anastomosis more difficult. Flap take-backs to theatre due to venous or arterial insufficiency were recorded in 14.0% of patients (42/300) and were successful in 55.8% of patients. This confirms lower rates of successful flap salvage in TMG flaps compared to DIEP flaps [24]. A potential reduction of fat necrosis in PAP flaps due to a more centralized perforator in the skin island is still discussed but exact perfusion studies to quantify the blood flow through the individual angiosome have not been conducted so far [22]. In an earlier study we observed 3.95% fat necrosis in patients receiving bilateral simultaneous reconstruction with DIEP flaps compared to 2.33% patients receiving TMG flaps [25]. Furthermore, a reduction of donor site seromas and consequential wound break down is proposed in PAP flaps since the incision is not as anterior compared to the TMG flap [22]. In our cohort of 300 unilateral breast reconstructions we observed 21 (7%) hematoseromas and 28 (9.3%). wound-healing disturbances. The observed donor site morbidity was comparable to reported DIEP flap and PAP flap studies [26,27]. Potential drawbacks of the PAP flap is a more challenging flap raise and possible repositioning of the patient if raised in prone position which leads to an increased operating time and potential other perioperative complications. Allen et al. started raising the PAP flap in a supine frog-leg position negating intraoperative repositioning. Hunter et al. followed this approach in their series and the modified supine lithotomy position is now widely accepted to avoid turning of the patient [22,28]. Another disadvantage is the potential absence of a reliable perforator in few cases and the recommendation of a preoperative Computer Tomography (CT)-angiogram beforehand [12,22]. In comparison the TMG flap does not need any preoperative diagnostic and perforators do not have to be verified using a doppler sonography intraoperatively. When the TMG flap is correctly raised and the skin island is left firmly attached to the underlying gracilis muscle, an estimated one to three perforators can reliably be expected [1].

It is also noteworthy that the majority of plastic surgeons is fairly familiar with the gracilis flap since it is widely used for facial nerve reanimation and extremity reconstruction alike. This fact and its constant and reliable anatomy consequently offer a shorter learning curve in breast reconstructions with the TMG compared to the PAP flap.

Dayan et al. came up with a diagonal skin island design of the TMG flap in 2013. In their small retrospective study of 10 flaps in nine patients they proposed that the so called Diagonal Upper Gracilis (DUG) flap allows greater recruitment of soft tissue with less tension on closure, hereby avoiding groin lymphatics and potential distortion of the gluteal fold [29]. From our most recent double center study including 300 patients we did not observe any signs of a post-operative lymphocele, and no patient needed a revision with closure of a lymphatic fistula. It is important to know however that the majority of the volume of the TMG flap should always be recruited in the posterior area of the thigh and an anterior dissection needs to be avoided to protect lymphatics. A comparable amount of hematoseromas was found in larger studies on PAP flaps where hematoseroma rate was 7.1% (18/265) compared to 21 (7%) donor site hematoseromas among 300 patients in this series [23]. The TMG flap’s skin island design is consistent with the widely adopted concept of planning incisions according to the relaxed skin tension lines, therefore reducing tension and wound break down as much as possible already. A diagonal design of the skin island is, therefore, not necessarily favored by the senior authors.

Nickl et al. propose a modified technique using only the small portion of the gracilis muscle underlying the skin island including the vascular pedicle when raising the TMG flap [30]. They argue that the amount of muscle atrophy over time can change the postoperative result and does not justify a potentially higher donor site morbidity. In our experience the muscle loss makes up about 50% of its cross-sectional area while it’s length remains almost the same. An estimate of 50% of the harvested muscle volume can be expected postoperatively which does justify its inclusion in the flap in our opinion, especially in slim patients who need every gram of volume for breast recontruction. Furthermore, the remaining muscle can serve as a recipient for secondary fat grafting. To our knowledge a donor site morbidity study which compares a large series of patients who had either minimal or maximal muscle harvest has not been conducted and the significant increase of the donor site morbidity has not been quantified yet.

Shaping techniques as described by other authors [30] as demonstrated in Figure 9 and Figure 10 were used to increase the flap projection and improve the overall shape of the breast after reconstruction. We dismissed these time consuming shaping techniques today since they do not seem to remain stable over time and flap inset can be more difficult using them in some instances due to the short pedicle. Therefore, we most often place the skin island in the lower pole and the muscle in the upper and lateral quadrant of the breast. The gracilis is then anchored to the pectoralis muscle using one or two strong resorbable sutures.

A noteworthy drawback of the TMG flap is the darker and sometimes mismatching colour of the skin island from the medial upper thigh which sometimes can include pubic hair. This can be an issue in secondary breast reconstructions in fair-skinned individuals and may need laser treatment postoperatively for hair removal and skin lightening/bleaching treatments [31].

In cases where larger flaps with skin islands greater than 10 × 30 cm and more than the average 250 to 300 g are needed for breast reconstruction an extended version of the TMG flap can be raised [19]. In few cases (*n* = 29) two TMG flaps were utilized for unilateral breast reconstruction by the senior authors when one flap did not offer enough volume [19]. Other authors described hybrid breast reconstructions using TMG flaps and a silicone implant instead when more volume and projection was needed [32].

In a study from 2018, the senior authors were able to demonstrate that the TMG flap offers a reliable scaffold for postoperative fat transfer if indicated. Here overall 83 out of 139 patients (59%) received postoperative Lipofilling to achieve better symmetry [33]. A similar percentage (54%; 163/300) of patients received lipofilling in the present study collective. In a study including 408 DIEP flaps and 56 LAP flaps, Opsomer et al. stated the need for later lipofilling to be 39% in LAP and 64% in their DIEP flaps [34]. The exact numbers regarding the indication of secondary surgeries including lipofilling are highly variable and depend on multiple factors though such as patient satisfaction, demand, and the individual local health care provider. A recent study demonstrated that increased age and overweight are no contraindications to breast reconstructions with the TMG flap and did not cause any increase of complications [35].

Compared to other popular and most commonly used alternatives to the DIEP flap for autologous breast reconstruction such as the lumbar artery perforator (LAP) flap, the superior gluteal artery perforator (SGAP) flap or the fasciocutaneous infragluteal free flap (FCI) flap, the TMG flap offers a comparable size and a similar complication profile, the mean pedicle length is significantly shorter though [36,37,38]. An overview of the common used free flaps and their detailed characteristics is shown in Table 5 [21,23,34,39,40].

Other observed donor site complications include a widening and caudal migration of the scar which can easily be corrected with scar revision and suture fixation to the deep fascia or the periost of the ischial tuberosity. A scar revision of the donor site with refixation of the gluteal fold was performed in 3/300 (1%) patients, as demonstrated in Figure 11, Figure 12, Figure 13, Figure 14 and Figure 15.

A complication sometimes observed after FCI flap harvest is post-operative cluneal nerve pain which was entirely absent in our series of 300 TMG flap donor sites [40]. This complication was observed in the early beginnings of the TMG flap raises by the senior authors but can be avoided when the soft tissue part of the flap posterior to the gracilis muscle is in a plane strictly superficial to the deep fascia. In the last three decades the TMG flap has been used for all types of breast reconstruction. It was used for uni- and bilateral primary, secondary and tertiary reconstructive procedures of the breast [6,11,41]. As described earlier, in rare instances where a larger unilateral breast reconstruction is anticipated two TMG flaps can be used when one flap is anastomosed to the mammary while the other the thoracodorsal artery and vein [42].

In massive weight loss patients who had a failed breast augmentation with implants or recurrent capsular fibrosis and ptosis the TMG flap represents a unique source for a bilateral autologous tertiary breast reconstruction while simultaneously addressing the excess skin in a similar fashion as a thigh lift procedure [4].

An important limitation of this study and the present literature in general is the fact that no patient reported outcomes (PROMs) after breast reconstructions with the TMG flap have been evaluated and published yet. More studies focussing on PROMs are needed for a meaningful and wholesome comparison of different free flaps used for breast reconstruction using the Breast-Q or similar questionnaires [43].

In summary, we have learned many lessons during the last 30 years of breast reconstructions using the TMG flap. Itemized key points are listed in Table 6 for a better overview. Safe and reliable uni-, and bilateral breast reconstructions with a low risk for donor site complications can be offered to patients using the TMG flap today. This flap rightly represents one of the most popular and widely offered alternatives to the DIEP flap. We hope that reconstructive surgeons worldwide will maintain this tradition and will implement the TMG flap into their armamentarium for breast reconstructions in the future, thereby offering a high and up to date standard of care and at the same time further improving the availability for breast reconstruction for all patients regardless of their body shape or body mass index and donor site characteristic.

## Figures and Tables

**Figure 1 jcm-10-03629-f001:**
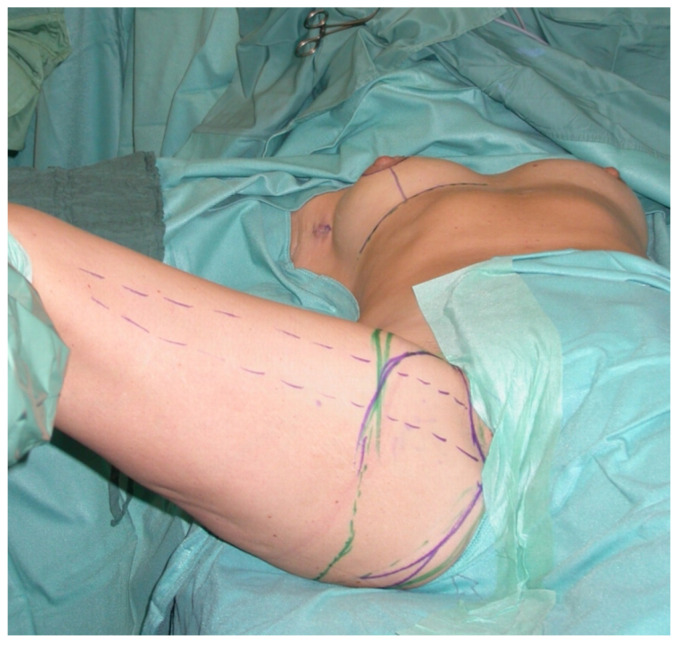
Intraoperative image at the beginning of the operation.

**Figure 2 jcm-10-03629-f002:**
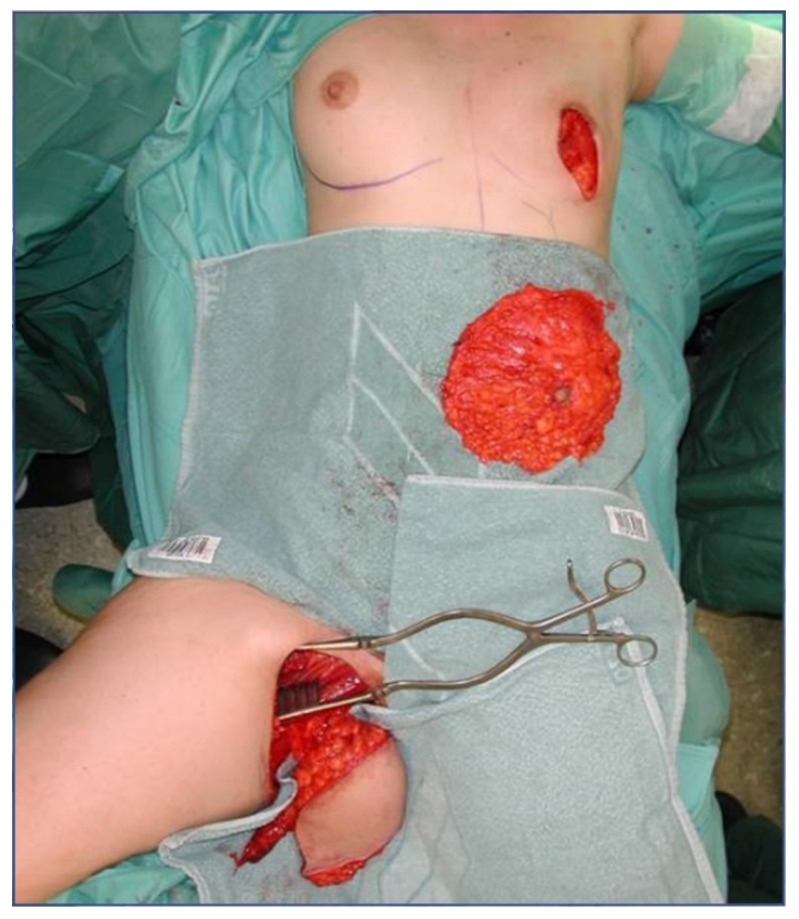
Intraoperative image after unilateral skin sparing mastectomy.

**Figure 3 jcm-10-03629-f003:**
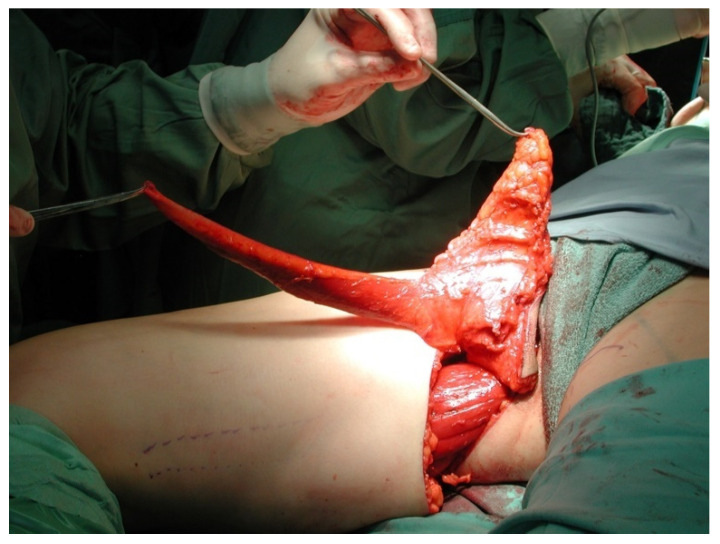
Intraoperative image after unilateral TMG flap raise.

**Figure 4 jcm-10-03629-f004:**
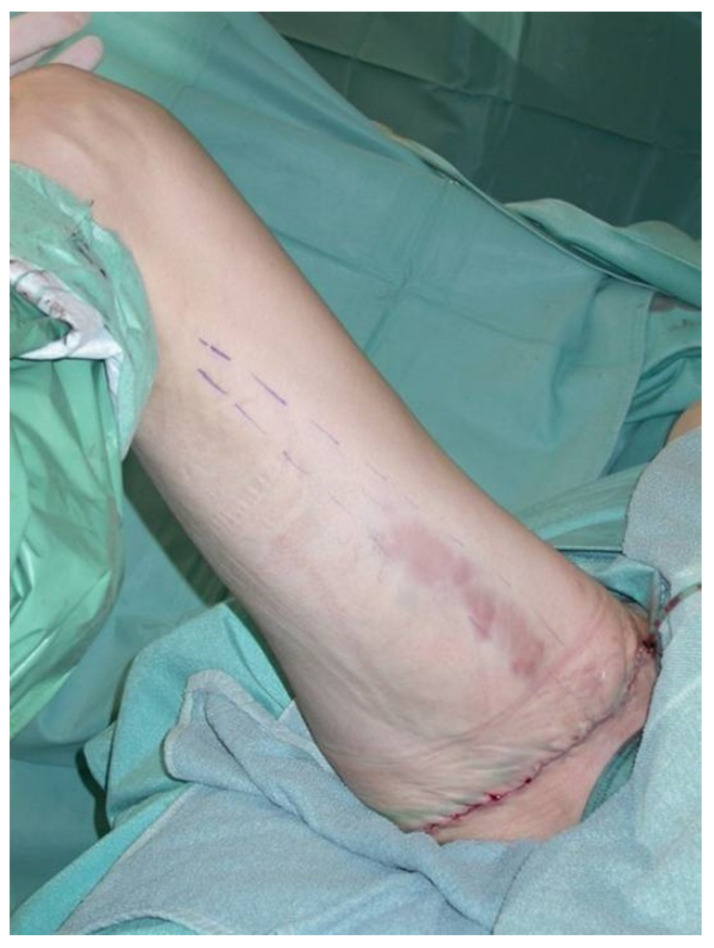
Intraoperative image after unilateral donor site closure.

**Figure 5 jcm-10-03629-f005:**
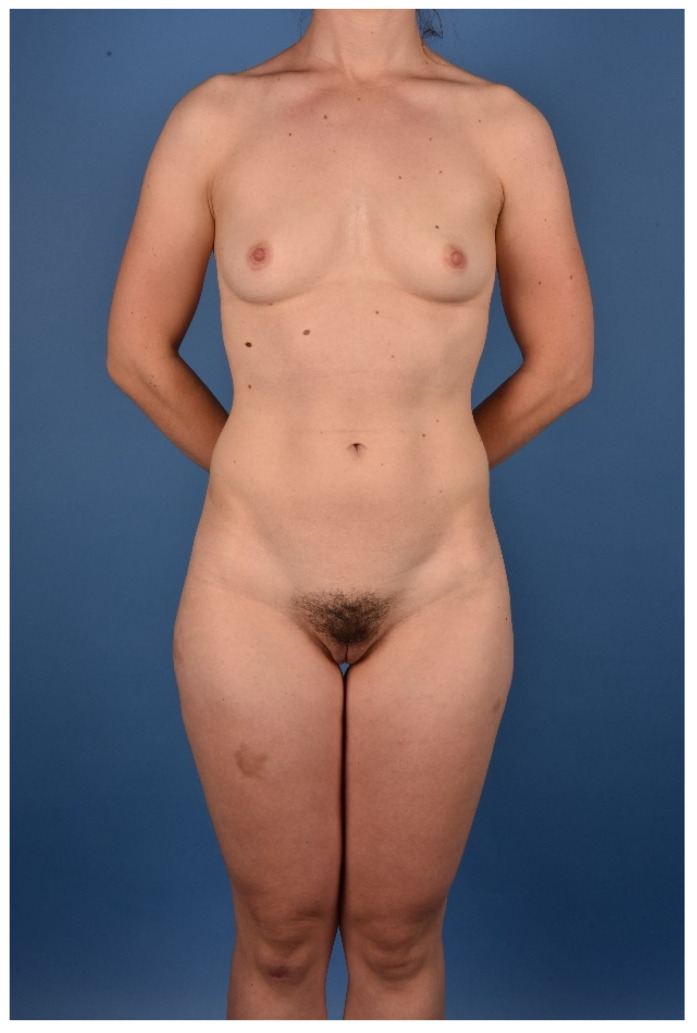
Preoperative image of a BRCA patient.

**Figure 6 jcm-10-03629-f006:**
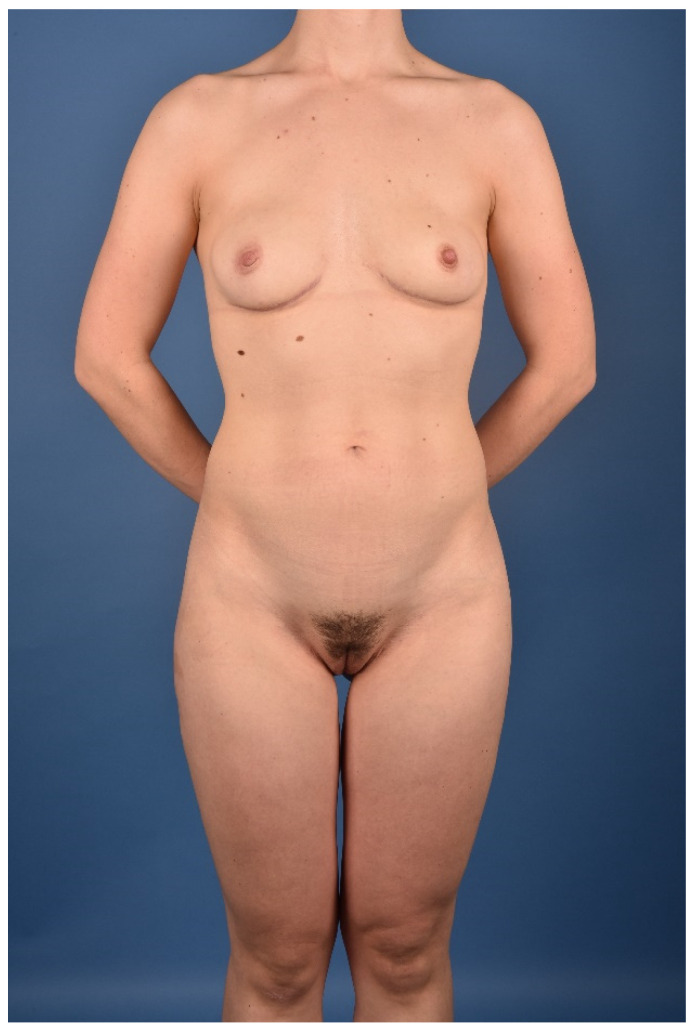
Postoperative image after two-staged bilateral autologous breast reconstruction with free TMG flaps.

**Figure 7 jcm-10-03629-f007:**
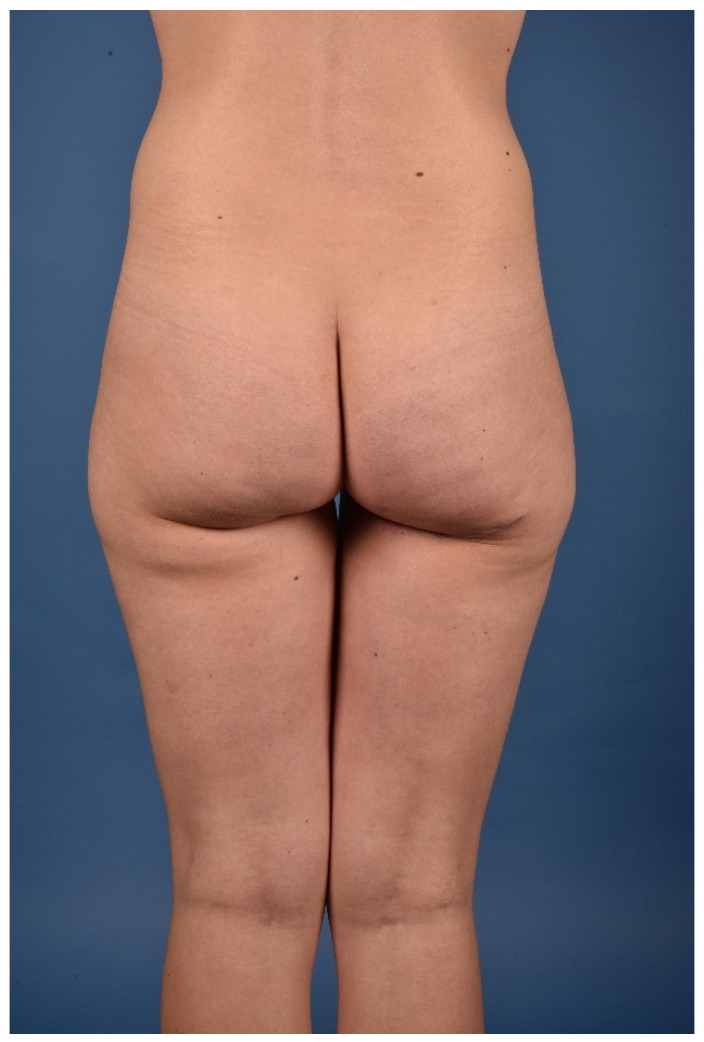
Postoperative donor site image.

**Figure 8 jcm-10-03629-f008:**
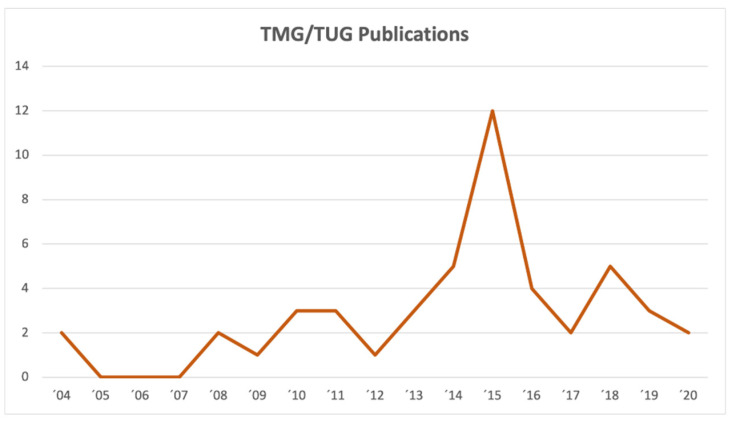
Transverse myocutaneous gracilis (TMG) flap publication trends between 2004 and today.

**Figure 9 jcm-10-03629-f009:**
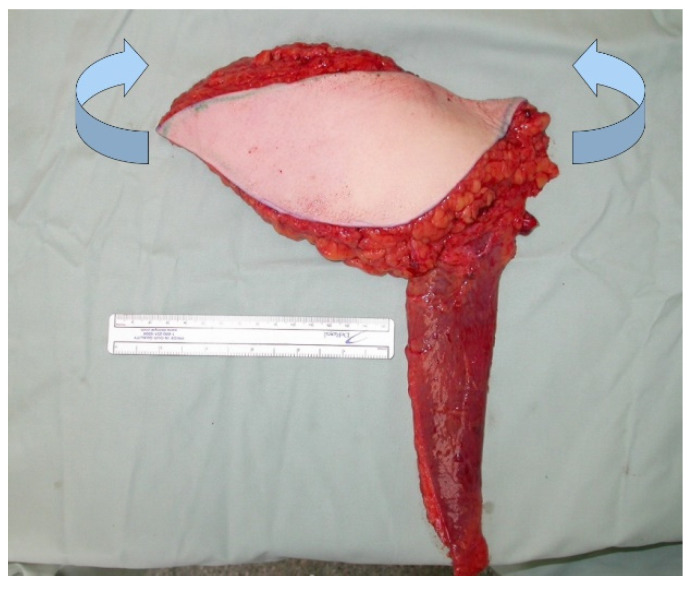
TMG flap before shaping (blue arrows indicate folding of the flap).

**Figure 10 jcm-10-03629-f010:**
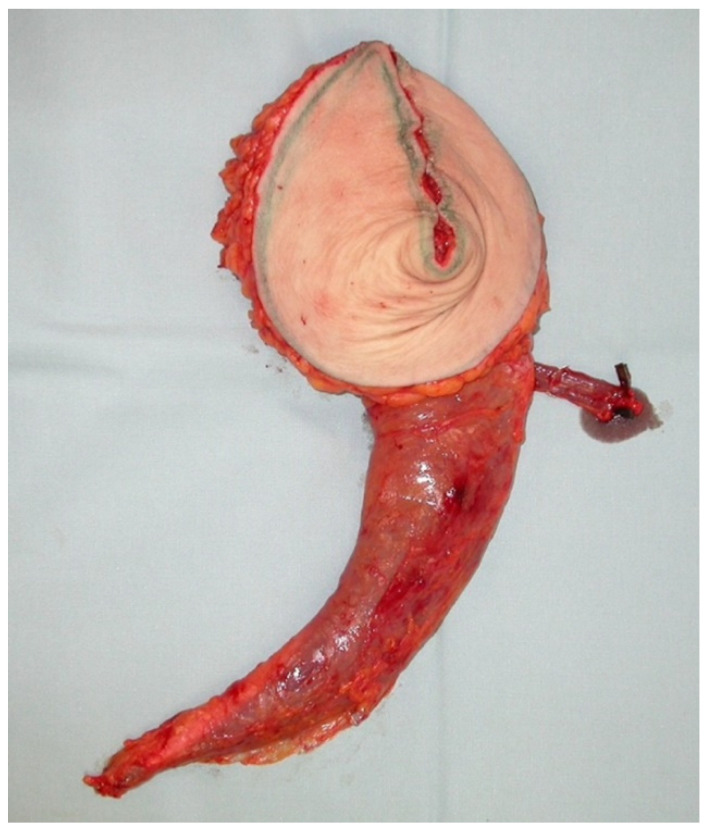
TMG flap after folding to generate an implant like flap shape.

**Figure 11 jcm-10-03629-f011:**
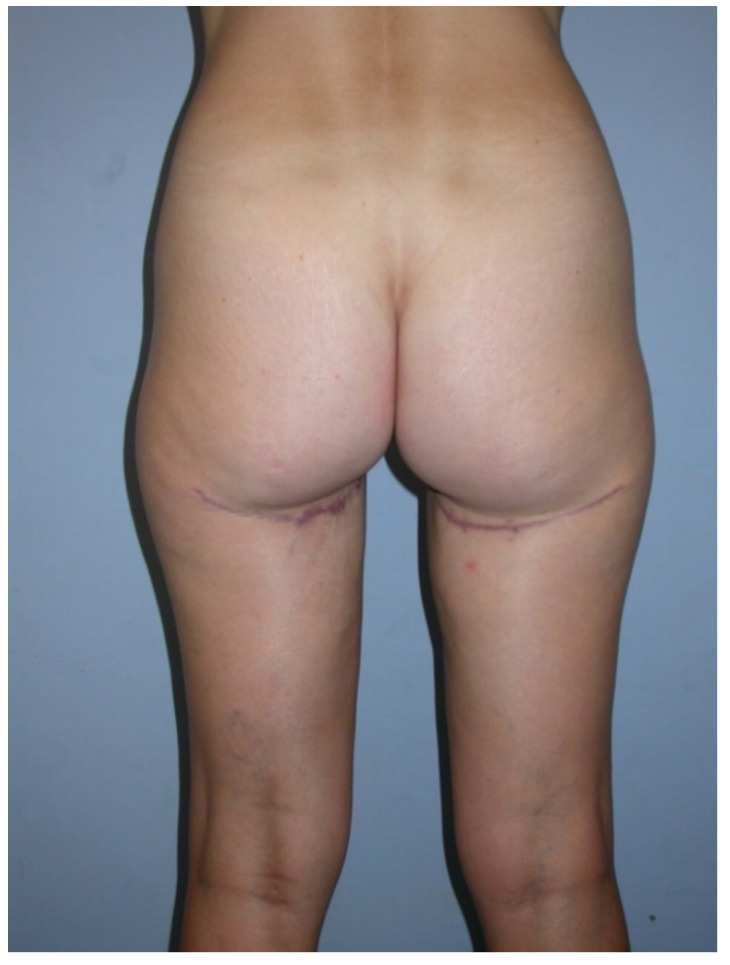
Post-operative caudal scar migration.

**Figure 12 jcm-10-03629-f012:**
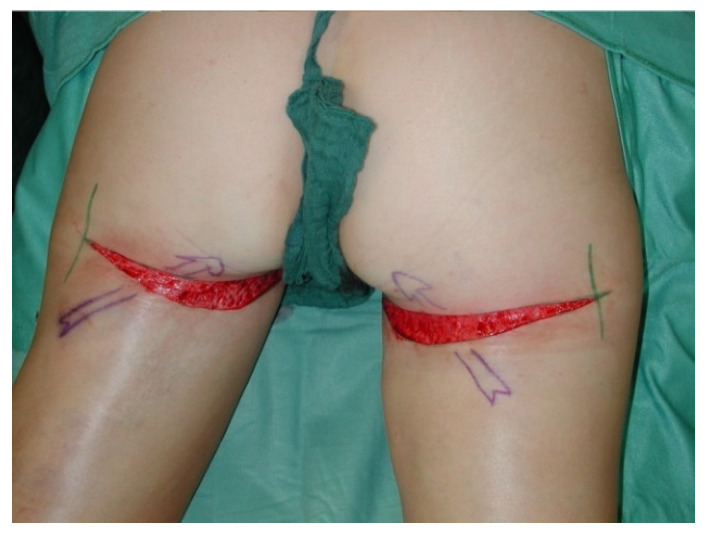
Deepithelialising of the scar for refixation.

**Figure 13 jcm-10-03629-f013:**
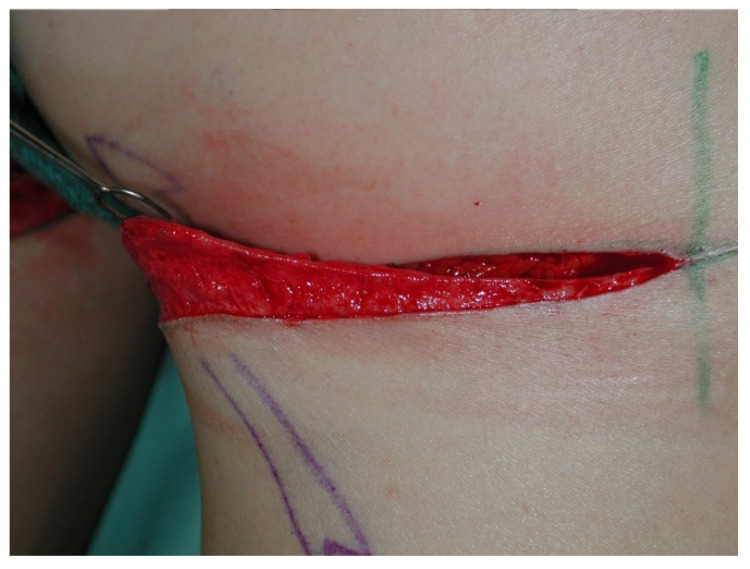
Underminig and anchoring to the ischial tuberosity or deep fascia using resorbable sutures.

**Figure 14 jcm-10-03629-f014:**
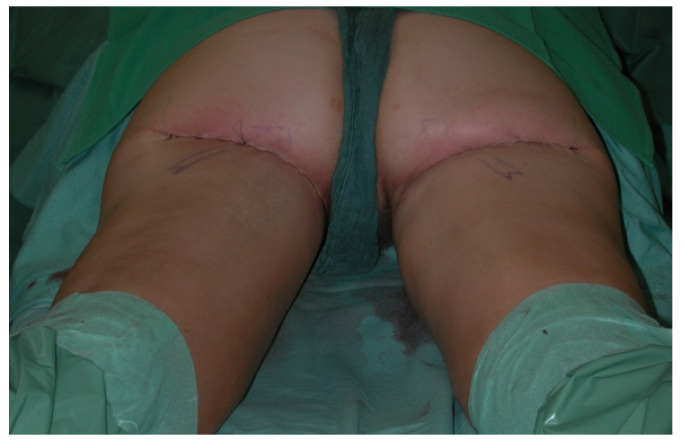
Skin closure after scar revision.

**Figure 15 jcm-10-03629-f015:**
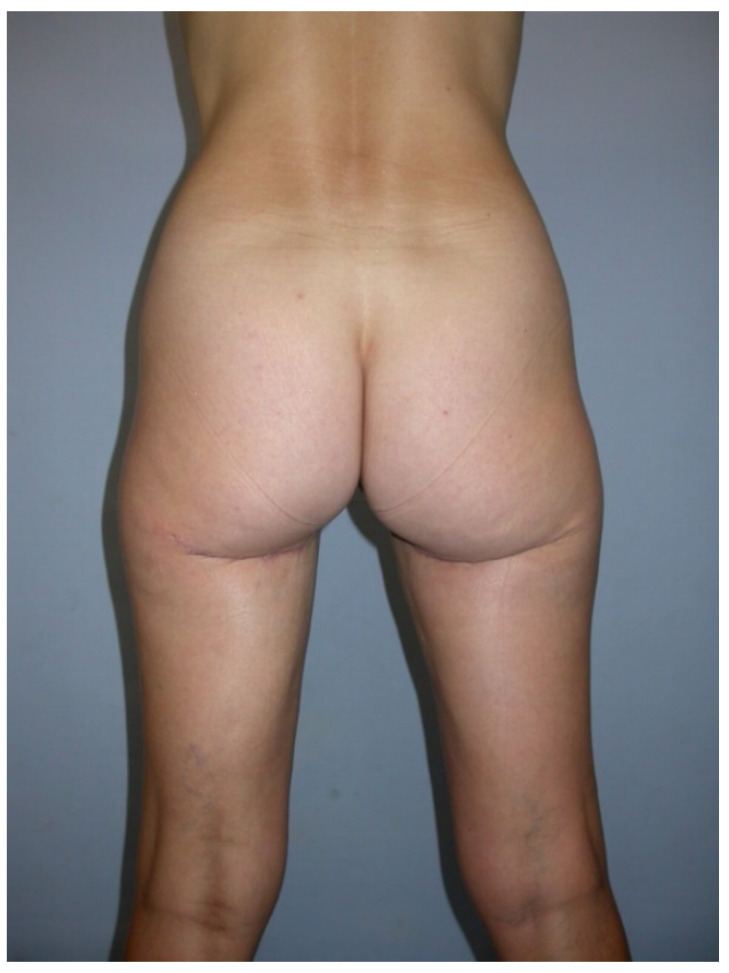
Post-operative result after scar revision with refixation of the gluteal fold.

**Table 1 jcm-10-03629-t001:** Classification of Complications (modified from Neaman et al. [14]).

**Major complications**	1. Hematoma or flap insufficiency requiring surgical intervention
	2. Seroma requiring aspiration or surgery
	3. Wound healing problems (also flap or fat necrosis) requiring surgery
	4. Cellulitis requiring iv antibiotics
	5. Deep Venous Thrombosis/Pulmonary Embolism
**Minor complications**	1. Hematoma without treatment (+erythrocyte substitution with no other treatment necessary)
	2. Seroma without treatment
	3. Delayed wound healing
	4. Cellulitis (also fat necrosis) treated with oral antibiotics without hospitalization

**Table 2 jcm-10-03629-t002:** Detailed patient characteristics.

Characteristic	Number		%
Cases included	300		100
Age, years			
Mean		48.0	
SD		10.6	
Follow-Up, months			
Mean		21.4	
SD		20.9	
Lipofilling	165		55.0
Preoperative radiotherapy	116		38.6
Type of reconstruction			
Primary	58		19.3
Secondary	124		41.3
Tertiary	118		39.4
Body Mass Index			
Mean		23.0	
SD		3.1	

**Table 3 jcm-10-03629-t003:** Flap characteristics.

Flap Characteristic	Mean	Range	*n*	%
Flap weight, gram	320	155–600		
Skin island diameter, centimeter				
width	9	7–13		
length	31	25–36		
Venous coupler, millimeter	2.5	1.5–3.5		
Pedicle length, millimeter	70	43–110		
Anastomosis				
Internal mammary (Artery /Vein)			276	92.0
Thoracodorsal (Artery/Vein)			24	8.0

**Table 4 jcm-10-03629-t004:** Complications overview; donor site, recipient site.

Complications	*n*	%
Overall	147	49.0
Major	83	27.0
Minor	70	23.0
Donor site complications		
Cellulitis	10	3.3
Hematoseroma	21	7.0
Wound healing disturbance	28	9.3
Scar revision	3	1.0
Recipient site complications		
Cellulitis	16	5.3
Hematoseroma	44	14.7
Wound healing disturbance	16	5.3
Fat necrosis	15	5.0
Flap take-back	42	14.0
Flap loss	19	6.3

**Table 5 jcm-10-03629-t005:** Overview of the common used free flaps for breast reconstruction and their properties.

Flap	Weight (Mean, gram)	Pedicle Length (Mean, millimeter)	Source (PubMed)
TMG	320 g	70 mm	Weitgasser et al., 2021
DIEP	550 g	150 mm	Blondeel et al., 1999 [38]
PAP	403 g	112 mm	Haddock et al., 2020 [23]
SGAP	400 g	85 mm	Zoccali et al., 2019 [39]
FCI	310 g	150 mm	Papp et al., 2011 [40]
LAP	499 g	40 mm	Opsomer et al., 2020 [37]

**Table 6 jcm-10-03629-t006:** Overview of key lessons learned in the last 30 years of TMG flap breast reconstruction.

**Flap characteristics**	320 g (155–600 g range)	Skin island diameter: 9 cm (7–13 cm range) in width and 31 cm (25 to 36 cm range) in length	Pedicle length: 70 mm (43 mm to 110 mm range)	
**Indications for surgery**	Qualified for primary, secondary and tertiary breast reconstructions	Can be used relatively independent of BMI and body shape		
**Anatomy**	Constant and reliable, no Computer Tomography (CT) Angio necessary for planning	Supercharging with the saphenous vein is possible but is rarely necessary	Flap is raised without repositioning	
**Flap shaping**	Do not offer long term form stability and can make flap inset more difficult	Are time consuming	Skin island is usually placed in the lower breast pole and the muscle in the upper pole	
**Donor site**	Lymphatic complications are uncommon	Donor site morbidity is comparable to other flaps	Dehiscence and wound break down is easy to manage, most often conservatively	Negative pressure wound therapy (NPWT) and skin grafting is rarely necessary
**Color missmatch**	Can be an issue in secondary reconstructions where local breast skin is replaced	Laser treatment and skin lightening or bleaching procedures can be offered		
**Widening of the donor site scar**	Not uncommon, especially in larger flaps and higher tenson on wound closure	Can easily be corrected by anchoring the revised scar to the ischial tuberosity or deep fascia		
**Cluneal nerve pain**	Occurs rarely and can be avoided through a dissection in a plane superficially to the deep fascia posterior to the gracilis			
**Large breast reconstructions**	Externded TMG flap can be used	Two flaps can be used for anastoming to the mammary and thoracodorsal vessels	Lipofilling procedures are a powerful tool for volume adjustment

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
