# Peer review of "Lessons Learned from 30 Years of Transverse Myocutaneous Gracilis Flap Breast Reconstruction: Historical Appraisal and Review of the Present Literature and 300 Cases"

_jcm, 2021, doi:10.3390/jcm10163629_

Round 1

Reviewer 1 Report

Thank you for your submission.

You mention that it is necessary to change position during surgery when raising the PAP flap. However this should not be the case as widely accepted and published by Hunter and Fahardi many years ago.(you even cite this publication). Why are you suggesting to include the whole of the gracilis muscle as this only adds to the donor site morbidity and does not add significant volume due to muscle atrophy over time. Also I would suggest to choose different pictures for your manuscript as the very well visible, rather unsightly donor sight scars should not be an example for junior plastic surgeons. You most probably agree that a well hidden scar within the gluteal fold, not extending too far lateral should be the goal when raising the TMG or PAP flap. Correctly you mention that the flap raising of the PAP is more challenging for less experienced surgeons. On the other hand the much shorter pedicle makes the microsurgical anastomosis and most of all any revisions much more challenging than the PAP flap. Other wise interesting work. Thank you again.

Author Response

Dear Editors, dear reviewer,

Thank you very much for your time and effort in reviewing our work. We appreciate your input and strongly believe it improves the overall quality of our manuscript. We hope the changes and improvements of our paper enable a publication. Please find the document explaining the conducted changes attached. 

Kind regards,

Laurenz Weitgasser

Reviewer 2 Report

The authors reviewed the literature and their 300 cases of breast reconstruction with TMG flaps and compared the results with literature findings of alternative free flap options.

The authors emphasized the lessons learned from 30 years of experience in the manuscript but still, they are not clear enough to appear even in the title. Itemized points of lessons should be described for clarity.

The authors included only the unilateral breast reconstruction cases for the retrospective review but figures used for the description of the surgical technique show the bilateral reconstruction setting. Figures for unilateral reconstruction need to be used here instead for better matching. 

The incidence of postoperative lipofilling, 55%, seems very high. This incidence should be compared with those of other popular alternative flaps in the literature. A higher chance of requiring lipofilling after breast reconstruction with a TMG flap can be a drawback if it is higher than others.

Volume information of lipofilling for the cohort should be disclosed.

The long mean length of stay, 12.4 days, needs to be explained. Is this related to TMG flap use? 

26% revision rate in table 2 appeared without further explanation. The revision should be defined and presented more specifically with classifications.

The flap loss rate written as 6.0% is incorrect. 19/300 gives 6.3%. 6.3% flap loss rate is now unacceptably high. This value should be compared with others’ and reasonably explained.

The study can be strengthened by comparing the authors’ series by a period as an evolution of the TMG flap use may be disclosed. It will be more informative if the authors point out what has changed and what has not based on the chronological comparison. This attempt can draw a real lesson learned.

Author Response

Dear Editors, dear reviewer,

Thank you very much for your time and effort in reviewing our work. We appreciate your input and strongly believe it improves the overall quality of our manuscript. We hope the changes and improvements of our paper enable a publication. Please find the conducted changes attached. 

Kind regards,

Laurenz Weitgasser
